

# Secondary organic aerosol formation from biomass burning emissions

Christopher Y. Lim[1], David H. Hagan[1], Matthew M. Coggon[2,3], Abigail R. Koss[2,3,4,a], Kanako Sekimoto[2,3,5], Joost de Gouw[2,4], Carsten Warneke[2,3], Christopher D. Cappa[6], Jesse H. Kroll[1]

[1]Department of Civil and Environmental Engineering, Massachusetts Institute of Technology, Cambridge, MA, USA
[2]Cooperative Institute for Research in Environmental Sciences, University of Colorado, Boulder, CO, USA
[3]NOAA Earth System Research Laboratory, Chemical Sciences Division, Boulder, CO, USA
[4]Department of Chemistry, University of Colorado, Boulder, CO, USA
[5]Graduate School of Nanobioscience, Yokohama City University, Yokohama, Japan
[6]Department of Civil and Environmental Engineering, University of California, Davis, CA, USA
[a]now at Department of Civil and Environmental Engineering, Massachusetts Institute of Technology, Cambridge, MA, USA

*Correspondence to*: Jesse H. Kroll (jhkroll@mit.edu)

**Abstract.** Biomass burning is an important source of aerosol and trace gases to the atmosphere, but how these emissions change chemically during their lifetimes is not fully understood. As part of the Fire Influence on Regional and Global
Environments Experiment (FIREX 2016), we investigated the effect of photochemical aging on biomass burning organic aerosol (BBOA), with a focus on fuels from the western United States. Emissions were sampled into a small (150 L) environmental chamber and photochemically aged via the addition of ozone and irradiation by 254 nm light. While some fraction of species undergoes photolysis, the vast majority of aging occurs via reaction with OH radicals, with total OH exposures corresponding to the equivalent of up to 10 days of atmospheric oxidation. For all fuels burned, large and rapid
changes are seen in the ensemble chemical composition of BBOA, as measured by an aerosol mass spectrometer (AMS). Secondary organic aerosol (SOA) formation is seen for all aging experiments and continues to grow with increasing OH exposure, but the magnitude of the SOA formation is highly variable between experiments. This variability can be explained well by a combination of experiment-to-experiment differences in OH exposure and the total concentration of non-methane organic gases (NMOGs) in the chamber before oxidation, measured by PTR-ToF-MS ($r^2$ values from 0.64 to 0.83). From this
relationship, we calculate the fraction of carbon from biomass burning NMOGs that is converted to SOA as a function of equivalent atmospheric aging time, with carbon yields ranging from $24 \pm 4$ % after 6 hours to $56 \pm 9$ % after 4 days.

## 1 Introduction

Biomass burning is a major source of particulate matter and trace gases to the atmosphere, and strongly affects global air quality and climate (Akagi et al., 2011; Bond et al., 2004; Liu et al., 2017). In fire-prone regions such as the western United
States, the frequency and intensity of wildfires have increased over the past several decades, due to fire management practices and climate change (Westerling et al., 2006), and this trend is expected to continue in the coming decades (Dennison et al., 2014; Spracklen et al., 2009). Emissions from fires have been the subject of intense study, but primary emissions alone do not



determine the atmospheric impacts of biomass burning, since smoke plumes can be transported thousands of kilometers and undergo dramatic chemical changes over their lifetimes in the atmosphere (Andreae et al., 1988; Cubison et al., 2011). In particular, biomass burning organic aerosol (BBOA) is subject to atmospheric aging processes that could significantly alter the climate- and health-relevant properties of biomass burning emissions (Hennigan et al., 2012; Vakkari et al., 2014). Such

processes include oxidation of gas-phase compounds followed by partitioning to the particle phase, forming secondary organic aerosol (SOA); direct oxidation of molecules in the particle phase through heterogeneous reactions; and evaporation of particulate semi-volatile molecules upon plume dilution, potentially followed by subsequent gas-phase oxidation. However, despite the potential importance of aging on biomass burning emissions, the effect of aging on BBOA composition and loading over multiday timescales is not well-constrained, and usually is not included in global chemical transport models (Shrivastava

et al., 2017).

Field measurements provide strong evidence that the composition of BBOA changes significantly when photochemically aged. In aircraft measurements of biomass burning plumes, OA consistently becomes more oxidized downwind, relative to the source of emissions (Capes et al., 2008; Cubison et al., 2011; Forrister et al., 2015; Jolleys et al., 2015; Jolleys et al., 2012).

Additionally, decreases in reactive tracers from biomass burning, such as levoglucosan, are observed after aging when compared to their contribution to fresh emissions (Cubison et al., 2011). Despite these consistencies, field measurements show mixed results with regards to whether or not there is an increase in net SOA downwind of fires. Net SOA formation is usually characterized by an OA enhancement ratio, defined as the ratio between fresh and aged $\Delta OA/\Delta CO$ measurements to account for plume dilution. Some studies show that little to no net secondary organic aerosol is formed over the course of several days

of aging, or even that a loss of organic mass can occur (Akagi et al., 2012; Capes et al., 2008; Cubison et al., 2011; Hecobian et al., 2011; Jolleys et al., 2015; Jolleys et al., 2012; May et al., 2015). However, other  studies show that significant OA enhancement can occur as well (DeCarlo et al., 2010; Vakkari et al., 2018; Yokelson et al., 2009).

Laboratory studies intended to constrain the effects of aging on biomass burning emissions have also had variable results.

Consistent with field measurements, laboratory experiments in which emissions from open burning and wood stoves were photochemically aged found that BBOA became increasingly oxidized and tracers were depleted with increased aging time (Ahern et al., 2019; Bertrand et al., 2018; Cubison et al., 2011; Grieshop et al., 2009; Hennigan et al., 2011; Ortega et al., 2013; Tkacik et al., 2017). Most laboratory experiments investigating the aging of biomass burning emissions find that significant amounts of SOA are formed in most, but not all, cases (Ahern et al., 2019; Bruns et al., 2016; Grieshop et al., 2009; Hennigan

et al., 2011; Ortega et al., 2013; Tiitta et al., 2016; Tkacik et al., 2017). Even under constrained laboratory experimental conditions, these studies show significant variability in SOA formation between burns of similar or even identical fuels. This variability is often attributed to differences in burning conditions (e.g., flaming and smoldering) (Hennigan et al., 2011) or the presence of unmeasured SOA precursors (Bruns et al., 2016; Grieshop et al., 2009; Ortega et al., 2013), but predicting biomass burning SOA across fuel types and burning conditions has remained a challenge. Very recently, Ahern et al. (2019) showed





that the detailed characterization of the hundreds of compounds emitted from a given burn, coupled with estimated SOA yields from each, enables the prediction of SOA formation to within roughly a factor of two. This approach establishes a clear link between the gas-phase emissions and SOA formation, but relies critically on a comprehensive understanding of emission profiles, which may exhibit substantial burn-to-burn variability.

The high degree of variability in net OA observed from biomass burning studies leads to a large range of estimates of SOA from biomass burning, which span nearly two orders of magnitude (Shrivastava et al., 2017). The range of global estimates is thus essentially unconstrained, with some studies ranking biomass burning as an insignificant source of SOA and others ranking it as the major source of global SOA (Shrivastava et al., 2015, 2017). Understanding the evolution of biomass burning

emissions is necessary to better evaluate the effects of biomass burning on air quality, human health, and climate. To this end, we describe the results from a set of laboratory aging experiments on a variety of fuels, employing an oxidation reactor coupled with real-time measurements of the composition of both the particle-phase and gas-phase emissions, to better constrain the effects of aging on biomass burning emissions.

## 2 Methods

### 2.1 Experimental setup and emissions sampling

Experiments were carried out as part of the Fire Influence on Regional and Global Environments Experiment (FIREX 2016) at the USDA Fire Sciences Laboratory (FSL) in Missoula, MT, with the goal of better understanding the evolution of biomass burning emissions within a controlled environment. Experiments took place during the "stack burn" portion of FIREX, in which fuels were burned beneath a 1.6 m diameter, 17 m tall exhaust stack, and were well mixed before being characterized

at the top of the stack. Fuels burned were characteristic of the western U.S. and included Engelmann spruce, lodgepole pine, subalpine fire, chamise, manzanita, Douglas fir, ponderosa pine, and sagebrush (Selimovic et al., 2018). For each of these fuels, components of each fuel (e.g., canopy, litter, duff) were burned individually, to determine differences between components, and in combination, as they would in natural wildfires and prescribed burns. In addition to these fuels, several other fuel types were included, such as peat, dung, Excelsior (wood shavings), rice straw, loblolly pine, and bear grass. The

weight of fuel used for each experiment was between 250–6000 g. Details of each burn sampled are given in the Supporting Information (SI Table 1); here we focus on results from 20 burns (out of 56 sampled total), for which aging experiments were carried out, the full analytical instrument suite (described in Sect. 2.2) collected data, and particle wall-loss rates were unaffected by UV irradiation (also discussed in Sect. 2.2).

Aging experiments were conducted in a 150 L PFA environmental chamber (the "mini-chamber"), an oxidation reactor of intermediate size between commonly-used oxidation flow reactors (with volumes of < 15 L) and large environmental chambers (> 1000 L). The mini-chamber was located in the wind tunnel room in the FSL; smoke from the top of the stack was transported





to the wind-tunnel room with a 30 m long community inlet (see Fig. S1). In order to minimize interactions of the smoke with the walls of the inlet, the tubing had a large diameter (8 in. diameter aluminum ducting) and fast flow (inline fan at the ducting exhaust pulling at 700 cfm, giving a transport time of < 2 s). Smoke from the community inlet was sub-sampled from the center of the flow using an ejector diluter pressurized with clean air, then passed through 1 m of passivated stainless-steel

tubing and a $PM_1$ cyclone, and injected into the chamber. Comparisons of fresh emissions between direct measurements from the top of the stack and measurements in the mini-chamber indicated some loss of gases and particles along the community inlet and transfer line, but these were relatively minor (< 8 % per volatility bin on average) and are not expected to affect results significantly (Fig. S2).

Prior to sampling, the chamber was flushed with clean air from a zero-air generator (Teledyne 701H) and humidified air (total 15 slpm) for approximately 45 minutes, leading to background particle concentrations $10^3$–$10^4$ times smaller than the peak concentrations during filling. Relative humidity remained in the range of 25–40 % throughout the entire experiment. Emissions sampling was initiated at the beginning of the burn and continued until the fuel was completely burned or the chamber particle concentration notably declined from its maximum concentration. Sampling generally lasted for a significant fraction of each

fire (5–20 minutes sampling time, while fires burned for 5–40 minutes) and initial dilution factors were approximately 7 times relative to concentrations in the stack. Once sampling was stopped, the chamber was allowed to mix passively for 5–10 minutes while the fresh emissions were characterized. The chamber was operated in semi-batch mode, meaning that after sampling, the chamber was continuously diluted with clean air while the smoke was oxidized and monitored. Chemical aging was initiated by OH, generated from both ozone photolysis and reaction of the resulting O($^1$D) with water vapor and the photolysis

of other OH precursors present in the smoke (e.g., HONO). A mercury pen-ray lamp (Jelight Model 600 ozone generator) was used to generate ozone (50–100 ppb in dilution air), which was added to the chamber starting just after the sampling period and continually added over the course of the experiment. The $O_3$ concentration in the chamber was typically lower at the start of each experiment and higher toward the end, and ranged from 10–80 ppb throughout. OH oxidation was initiated by exposing the chamber to 254 nm UV light (one UVP, LLC. XX-40S bulb, for a chamber averaged photon flux of ~3 x $10^{15}$ photons cm$^{-}$

$^2$ s$^{-1}$). Use of low-wavelength UV light can introduce non-OH chemistry (Peng et al., 2016); however, loss rates of common compounds from biomass burning such as toluene, phenol, and naphthalene agree well with predicted loss rates from OH reaction alone (average [OH] $\approx$ 2 x $10^8$ molec cm$^{-3}$), indicating that for these compounds UV photolysis is negligible. However, photolysis can be competitive with OH reactions for compounds that undergo rapid photolytic degradation at 254 nm. Such species, which are characterized by a low ratio between their OH reaction rate constant to absorption cross section ($k_{OH}$ / $\sigma_{254nm}$

< 1.5 x $10^7$ cm/s), and high quantum yields, include conjugated carbonyls (e.g., furfural, benzaldehyde) (Coggon et al., in preparation). These species will therefore undergo substantially more photolysis in the present experiments compared to atmospheric conditions; however, for the vast majority of compounds studied here, the dominant reactive loss (as in the atmosphere) is by reaction with OH (Coggon et al., in preparation). Oxidation lasted 30–60 minutes, then the chamber was flushed with clean air before the next experiment. At the end of each day, the chamber was left to flush with clean air and with



four UV lamps on overnight. In addition to the aging experiments, control runs were conducted without UV light and used to characterize the evolution of the smoke in the absence of OH oxidation.

Particles and gases exiting the reactor were monitored with a suite of analytical instrumentation. Particle composition measurements were made with an aerosol mass spectrometer with a standard tungsten vaporizer (AMS, Aerodyne Research, Inc.) which measures the mass and composition of non-refractory particles with diameters between 70 nm and 1 μm (DeCarlo et al., 2006). Black carbon mass was measured with a Single Particle Soot Photometer (SP2, Droplet Measurement Technologies) and SP-AMS (Aerodyne Research Inc., Onasch et al., (2012)). Particle size distributions were measured with a Scanning Electrical Mobility Spectrometer (SEMS, Brechtel). All particle-phase measurements were made alternating between a two-stage thermal denuder (150 °C and 250 °C) and a room-temperature bypass line. From these measurements, particle organic mass fraction remaining (MFR) was calculated by comparing the thermally denuded particle organic mass to the mass after the bypass line. Non-methane organic gases (NMOGs) were measured with a proton-transfer-reaction time-of-flight mass spectrometer (NOAA PTR-ToF-MS); these measurements are described in detail elsewhere (Koss et al., 2018, Sekimoto et al., 2018), and the oxidation chemistry of NMOGs is described in a companion paper (Coggon et al., in preparation). Auxiliary measurements of inorganic gases included $O_3$ (2B Technologies, Model 202), CO (Teledyne, Model T300), $CO_2$ (LI-COR, LI-840A), and $SO_4/SO_2$ (Thermo Fisher 5020i). Particle optical properties were monitored with a three-wavelength photoacoustic spectrometer (PASS-3), a two-wavelength cavity ring down photoacoustic spectrometer (CRD-PAS), and a cavity attenuated phase shift spectrometer (CAPS). Optical measurements from these instruments are not used in the present study, and instead will be discussed in a future publication (Cappa et al., in preparation).

## 2.2 Data analysis

Particle mass and composition data from the AMS were analyzed using the ToF-AMS analysis toolkits (Squirrel version 1.57I, Pika version 1.16I) using the "improved-ambient" method for calculating oxygen-to-carbon (O/C) and hydrogen-to-carbon (H/C) elemental ratios (Canagaratna et al., 2015). Mass concentrations must be corrected for changes in AMS collection efficiency (CE), as well as wall loss and dilution. This is often done in chamber experiments by normalizing measured SOA mass concentrations to that of an inert internal tracer. However, we were unable to find a suitable tracer in these experiments: sulfate changes as a result of oxidation of emitted $SO_2$, black carbon (when present in high concentrations) exhibited wall losses different from OA (as described below), and POA tracers (such as the $C_7H_{11}^+$ ion, recently used by Ahern et al. (2019)) are likely to be lost via heterogeneous oxidation at the high OH exposures examined here. Thus, corrections for CE, dilution, and particle wall loss were carried out individually, as described below.

CE and particle density were calculated by comparing AMS particle time-of-flight (PToF) and SEMS size distributions (Bahreini et al., 2005) for a subset of data points with sufficient PToF signal. An AMS CE correction was then applied to the entire data set by parameterizing the exponential relationship between AMS CE with the particle MFR (Fig. S3). Generally,





particle MFR increases with increased oxidation, indicating that particles are becoming less volatile and more likely to be (semi-)solid, and therefore likely to have a lower CE (due to increased bounce off of the AMS vaporizer). Calculated AMS collection efficiencies range from 0.35 to 0.64, with the average CE of fresh emissions equal to 0.54 and average CE of aged (i.e., end-of-oxidation) emissions equal to 0.40. We note that this CE value of 0.54 for fresh emissions is substantially lower than the value of 1 typically assumed in such experiments (Ahern et al., 2019; Hennigan et al., 2011; Heringa et al., 2011; Ortega et al., 2013; Tkacik et al., 2017).

Acetonitrile, an inert tracer species ($k_{OH}$ = 2.16 x 10$^{-14}$ cm$^3$ molec$^{-1}$ s$^{-1}$) (Atkinson et al., 2001), was used to correct all data for chamber dilution. An exponential function was fitted to the decay of acetonitrile for each experiment ($\tau_{dilution}$ ~ 20 minutes). All gas- and particle-phase species concentrations are corrected for dilution using an experiment-specific dilution rate. Dilution rates calculated from the fitted decay of acetonitrile agree well with estimates based on chamber volume and flow rates; this is in contrast with dilution rates derived from CO, which is produced in the chamber during oxidation. Particle mass concentrations were also corrected for loss to the chamber walls, by fitting an exponential equation to the particle organic mass concentration during a control experiment in which the UV lamps were not turned on ($\tau_{wall}$ = 35 min; Fig. S4). Comparison of wall loss rates calculated between two dark experiments and size-dependent wall loss rates calculated from SEMS data yield similar results. Fires with the lowest initial AMS OA to SP-AMS BC ratios (OA/rBC < 3.4) showed enhanced wall loss rates upon UV irradiation, likely due to photoionization of BC or polycyclic aromatic hydrocarbons present on the particle surface (Burtscher, 1992; Mohr et al., 1996), followed by increased electrostatic interactions with the chamber walls. This complicates the wall loss correction, and such low OA/rBC burns are excluded from this analysis (see SI Table 1). All particle concentrations presented, unless otherwise noted, are corrected for collection efficiency, dilution, and wall loss. Gases were corrected for dilution only; control runs showed that wall loss was not a major loss pathway for most primary intermediate volatility organic compounds (IVOCs; Fig. S5).

OH exposures in the chamber were estimated by measuring the decay of an OH tracer, deuterated *n*-butanol (D9, 98%, Cambridge Isotope Laboratories), added at the beginning of each experiment (10 μL, 2% in water). OH exposure was then calculated from the dilution-corrected concentration of deuterated butanol and its reaction rate coefficient with OH ($k$ = 3.4 x 10$^{-12}$ cm$^3$ molec$^{-1}$ s$^{-1}$) (Barmet et al., 2012). Differences in the initial concentration of OH precursors (e.g., O$_3$ and HONO) in the chamber, OH sinks, and experiment duration resulted in variations in total OH exposure from experiment to experiment, with end-of-experiment OH exposures (calculated at the point when UV lamps are turned off) ranging from 1–10 days of atmospheric aging. Throughout this work, OH exposure is converted to equivalent atmospheric aging time (in days) by assuming an average OH concentration of 1.5 x 10$^6$ molec cm$^{-3}$.



## 3 Results and discussion

### 3.1 Loading and composition of fresh and aged biomass burning particles

The total, initial aerosol mass in the chamber varied widely from experiment to experiment, averaging $130 \pm 103$ µg m$^{-3}$ (mean $\pm 1\sigma$), depending on the amount of fuel burned, fuel type, sampling time, and dilution factor prior to oxidation (SI Table 1).

For all experiments considered here, the organic fraction dominated the composition of the primary particle mass (as measured by the sum total of the AMS and SP2, with initial primary OA (POA) concentrations accounting for 70–99 % of the total aerosol mass. The initial fraction of black carbon (BC) mass also varied significantly (0–30 % of total aerosol mass) and was highly dependent on fuel type, with the highest BC mass fractions observed for chaparral and Excelsior fuels. Concentrations of non-BC inorganic components (AMS measured nitrate, sulfate, ammonium, and chloride) were variable, but low for all

experiments ($\leq 8$ % of total aerosol mass). After initiation of oxidation, organic aerosol loadings grew substantially for all experiments (Fig. 1a). The average mass loading of SOA formed (corrected OA mass at the end of the experiment, minus the OA mass prior to OH oxidation) was $260 \pm 250$ µg m$^{-3}$. Alternatively, the amount of SOA at the end of experiment can be expressed as an OA enhancement ratio, defined as the final OA mass (secondary + primary) divided by the initial (primary) OA mass. The average OA enhancement ratio was $3.5 \pm 1.7$. This is considerably higher than reported in previous studies

(Ahern et al., 2019; Hennigan et al., 2011; Ortega et al., 2016; Tkacik et al., 2017), but once differences in OH exposure (as well as AMS CE) are taken into account, these results are broadly consistent with previous studies (Fig. S6). Discussion of the relationship between SOA formation and gas phase composition will be discussed further in Sect. 3.2.

The chemical composition of the primary organic particulate matter varied significantly between experiments. The initial O/C

ranged from 0.20 to 0.60 ($0.35 \pm 0.09$) and initial H/C ranged from 1.72 to 1.85 ($1.77 \pm 0.04$), indicating significant variation in POA elemental composition. Elemental ratios stayed constant in the chamber for each given experiment until the lights were turned on. Similarly, control experiments (with no oxidation) showed constant OA/BC, O/C, and H/C over the course of the run, indicating that OA mass and overall oxidation state of biomass burning POA is stable in the chamber without exposure to UV light, despite the potential for semi-volatile compounds to partition to the gas phase and/or be lost to the chamber walls

(Ahern et al., 2019; Grieshop et al., 2009; May et al., 2015).

OH oxidation, initiated in the chamber by exposure to UV light, rapidly changes the composition of OA, as shown in Fig. 1b–d. Gas phase chemistry in the mini-chamber will be discussed in detail in a forthcoming publication (Coggon et al., in preparation). Despite differences in initial composition between experiments, OA in all experiments undergoes a large increase

in O/C and decrease in H/C (Fig. 1b), and a corresponding increase in average carbon oxidation state (OS$_C$, equal to 2 O/C – H/C) (Fig. 1c). OS$_C$ for all fuels increases with increasing OH exposure, with an average, end-of-experiment increase in OS$_C$ of $1.33 \pm 0.50$. Most of this change occurs during the initial period of oxidation (equivalent timescales of just 1–2 days); after this, changes in OS$_C$ still occur, but over much longer timescales. Both condensation of SOA and heterogeneous oxidation





(direct reactions between gas-phase oxidants and particle-phase organic molecules) can contribute to the observed increases in oxidation state. However, the initial rate of change for aging of biomass burning emissions (gases and particles combined) is much faster than the average rate of $OS_C$ change measured in laboratory heterogeneous OH-oxidation experiments (Kroll et al., 2015) (shown as the grey line in Fig. 1c), implying that condensation of highly oxidized secondary mass is the main driver

for the changes in composition observed.

Similar to the elemental ratios, the initial fraction of the primary organic signal from the AMS fragment ion $C_2H_4O_2^+$ ($f_{C2H4O2+}$) varies from burn to burn (mean 2.2 ± 1.2 %). This fragment is a small (< 6 %) contributor to the overall OA mass spectrum, but is known to correlate with levoglucosan (and related molecules) and is commonly used as a tracer for biomass burning

POA (Cubison et al., 2011). Figure 1d shows the evolution of $f_{C2H4O2+}$, normalized to its value at the start of each experiment. This ion is known to correspond to semi-volatile species (Grieshop et al., 2009), and changes to $f_{C2H4O2+}$ (~ 25 % loss) are observed even in the absence of oxidation; however, oxidation greatly enhances the rate and magnitude of its decrease. Under oxidation conditions, the contribution of this fragment to the total organic mass decreases dramatically over the course of 1–2 days of equivalent OH exposure, then stabilizes after that. This is in agreement with previous aircraft studies, which show that

even in highly aged airmasses, a small amount of this tracer remains elevated relative to the atmospheric background level (Cubison et al., 2011), potentially due to contribution of other molecules to this tracer. Overall, the chemical changes observed here are likely dominated by the formation of SOA, although heterogeneous oxidation, dilution-driven evaporation, and wall loss may also contribute.

**3.2 Secondary organic aerosol formation**

All aging experiments show substantial SOA formation, with OA mass continuing to increase with extended aging time (Fig. 1a). Consistent with previous studies, the correlations between OA mass enhancement ratio and various parameters that could affect SOA production (e.g., OH exposure, POA, common SOA precursors, total NMOG concentration) are weak at best (Fig. S7). However, the absolute amount of SOA formed does appear to correlate with some of these. Figure 2 shows the relationship between SOA formed by the end of each experiment (μg m$^{-3}$) and the initial concentration of total NMOGs in the chamber

(ppb) measured by the PTR-ToF-MS. A positive correlation is seen between the two variables, but the correlation is not especially strong ($r^2$ = 0.51). A confounding variable in this relationship is the difference in total OH exposure between experiments, as shown in the color scale. To account for the differences in OH exposure, Fig. 3 shows the same relationship between SOA and NMOG concentration, but now compared at equal OH exposures (0.25, 0.5, 1, 2, 3, and 4 days of atmospheric aging) and with both axes converted to carbon concentration. Although there is still significant scatter, possibly

due to differences in initial POA levels (Fig. S8), there is an improved correlation ($r^2$ = 0.64–0.83) between SOA and NMOG carbon mass at each OH exposure. Subplots in Fig. 3 contain different numbers of data points, due to the differences in final OH exposure achieved in each experiment. As such, the $r^2$ values are not strictly comparable, but are labeled to show the correlation in all cases. Additionally, the slopes of these plots exhibit a clear trend, increasing with OH exposure. This indicates



that a greater fraction of carbon from NMOGs is converted to SOA as aging time increases, consistent with continual SOA formation over long aging timescales. Total PTR-ToF-MS measured NMOGs also correlates well with POA concentration; as such, POA shows a similarly strong relationship with SOA (Fig. S9).

The correlation between SOA formation and the initial chamber concentration of NMOGs is reasonable, since NMOGs provide the carbon that drives SOA growth. However, the PTR-ToF-MS measures many compounds that likely do not contribute to SOA formation (e.g., small compounds such as methanol and acetonitrile). In addition to comparing SOA to the initial NMOG concentration, we can examine how SOA formation correlates with the concentration of NMOGs above some molecular weight cutoff. Figure 4 shows the correlation coefficient for the linear fit between SOA carbon mass and summed NMOG carbon

mass at each molecular weight cutoff for 1 day of equivalent aging. While low molecular weight NMOGs are not expected to contribute to SOA mass, the correlation between SOA carbon mass and initial NMOG carbon mass does not improve substantially when these are excluded (left side of Fig. 4). In fact, the correlation between SOA is relatively insensitive to the $m/z$ cutoff point, until only compounds with molecular weight greater than monoterpenes ($m/z > 137$) are considered. After this point the correlation drops rapidly to zero, likely because the PTR-ToF-MS signal is very low in this mass range, and/or

compounds with 10 or fewer carbon atoms are major contributors to SOA formation (or correlate with some unmeasured SOA-forming species). This suggests that the ratio of SOA precursors to the total concentration of measured NMOGs is relatively constant between experiments. The rapid step change after $m/z$ 137 might suggest the importance of monoterpene chemistry to SOA formation, consistent with Ahern et al. (2019); however, we do not observe strong correlations between SOA and monoterpenes alone, nor any other single SOA precursor (e.g., isoprene), nor any subset or class of compounds measured by

the PTR-ToF-MS (e.g., IVOCs). PTR-ToF-MS measurements taken directly from the FSL exhaust stack show that although the NMOG emissions are incredibly complex, much of the variability in emissions (~85 %) can be described by just two emission factors (derived using positive matrix factorization, or PMF), one high-temperature combustion factor and one low-temperature combustion factor (Sekimoto et al., 2018). We do not see improved correlations between NMOGs and SOA when splitting total NMOGs by factor type, indicating that both factors contain compounds that contribute to SOA formation.


Previous biomass burning aging experiments with both aerosol and NMOG measurements have not observed this relationship between SOA and total NMOGs (Ortega et al., 2013) or individual SOA precursors (Bruns et al., 2016; Grieshop et al., 2009; Ortega et al., 2013; Tkacik et al., 2017). Most such studies identified only half or less of the NMOG signal and/or were limited to a small number of experiments (Grieshop et al., 2009; Ortega et al., 2013; Tkacik et al., 2017); this could potentially explain

why similar correlations between total VOCs and SOA have not been observed before. By contrast, the use of PTR-ToF-MS in the present study enables the measurement of ~50–80 % of reactive gas-phase carbon from biomass burning, making it a good tool for characterizing the total organic gas-phase emissions from fires (Hatch et al., 2017). In this analysis, we have included both identified and calibrated PTR-ToF-MS ions (~90 % of the signal, approximately 150 ions) as well as an additional 370 unidentified ions (Koss et al., 2018). This is roughly comparable to the number of compounds identified and



quantified in recent speciated studies of biomass burning emissions (Hatch et al., 2015), which have been shown to enable reasonable bottom-up estimates of total SOA formation (Ahern et al., 2019). However, such speciated approaches require estimates of SOA yields from each precursor, which might be unknown or highly uncertain. As an alternative approach, the initial total NMOG concentration measured by the PTR-ToF-MS, in conjunction with OH exposure, provides a reasonable

predictor for the amount of SOA formation without the need for speciated aerosol yields.

Recent work has pointed to the importance of "non-traditional" SOA precursors to SOA formation for residential wood combustion of a single fuel type (beech wood) (Bruns et al., 2016). These precursors include semivolatile and intermediate-volatility volatile organic compounds (S/IVOCs) such as phenols and naphthalenes (Bruns et al., 2016). However, the majority

of gas-phase carbon observed in this study is in compounds with low carbon number ($n_C < 7$), with corresponding volatilities that are weighted towards volatile compounds ($c_0 > 10^7$ µg m$^{-3}$) rather than S/IVOCs (Fig. 5). This means that SOA from biomass burning is strongly influenced by the oxidation of relatively small, volatile species, and not S/IVOCs, a result consistent with Ahern et al. (2019); alternatively, this could mean that the PTR-ToF-MS does not measure all important SOA precursors, but measures compounds that are co-emitted and correlate well with them.

From each of the relationships between SOA carbon mass and initial NMOG carbon mass (linear fits in Fig. 3), an effective carbon yield can be calculated. Carbon yield is defined here as the SOA formed at a given OH exposure divided by the total NMOG carbon reacted at each respective OH exposure. The amount of gas phase carbon reacted ($\Delta[C]_{NMOG}$) is estimated from the initial concentration of PTR-ToF-MS measured gas-phase organic carbon in the chamber before oxidation, experiment-

specific dilution rates, and speciated OH reaction rates for identified compounds present in each fire (Koss et al., 2018). Carbon yields as a function of atmospheric age are shown in Fig. 6, and range from $24 \pm 4$ % at 0.25 days of equivalent atmospheric oxidation to $56 \pm 9$ % after 4 days of equivalent atmospheric oxidation. Since the PTR-ToF-MS does not measure the true total NMOG concentration (e.g., not all alkanes and alkenes are measured), these carbon yields are likely to be upper bounds. The calculated yields use our best estimate of OA carbon mass, using the AMS CE correction described previously. The grey points

in Fig. 6 show the carbon yields assuming a constant AMS CE equal to 1. In both cases, the amount of SOA formed increases with increasing aging time. Accounting for changes in collection efficiency that occur with aging leads to an approximate factor of 4 increase in yields; previous studies assuming a constant or initial AMS CE equal to 1 may significantly underestimate SOA formation (Ahern et al., 2019; Hennigan et al., 2011; Heringa et al., 2011; Ortega et al., 2013). Better constraints on estimates of AMS collection efficiency are needed for improved estimates of SOA formation from biomass

burning. Nonetheless, this carbon yield, combined with laboratory- or field-based estimates of NMOG emissions, provides a means for including SOA formation from biomass burning sources within chemical transport models.



The variability in findings from previous lab and field studies on the effect aging has on net SOA from biomass burning can be potentially explained by the effects of dilution on the evolution of BBOA mass. Some fraction of BBOA is semi-volatile, and dilution (in chambers or ambient smoke plumes) will cause volatile OA components to partition from the particle phase to the gas phase (May et al., 2013). Recent modeling work has shown that even in plumes that show no net SOA formation,

significant condensation of secondary organic mass may occur  (Bian et al., 2017), but net growth can be low (or even negligible) due to dilution-driven evaporation of OA. In ambient plumes, dilution drives semi-volatile species from the particle to gas phase; although this causes a loss in OA mass, it also serves as a source of SVOCs that can condense back onto particles after oxidation, leading to little to no net change in OA. Related to this point, calculated net OA values are also sensitive to the choice of starting point (i.e., $t_0$). Initial dilution in chamber experiments may result in substantial POA evaporation, which

provide high concentrations of SVOCs that are efficiently converted to SOA upon oxidation. However, some laboratory experiments find a net loss of OA mass during aging (Hennigan et al., 2011; Ortega et al., 2013; Tkacik et al., 2017), which we do not observe in the present experiments. Instead we see SOA formation in all photooxidation studies, and no evaporative loss during dilution-only experiments, results consistent with recent work by Ahern et al. ( 2019). The reason for this is unclear, but could be due to some combination of differences in dilution and wall losses (gas and particle) in chambers and flow tubes,

the method of AMS collection efficiency correction, and experimental conditions leading to different peroxy radical ($RO_2$) chemistry. Chambers such as the mini-chamber have lower initial dilution factors, but much higher OH concentrations, potentially favoring condensation from VOC oxidation over evaporation of particle mass. With the potential preference for condensation over evaporation, this present study may be effectively measuring the potential SOA formation while excluding evaporation from the extensive dilution that occurs in biomass burning plumes; therefore, it is primarily accessing the

"chemistry" component of OA evolution (Bian et al., 2017) Thus, the carbon yields shown in Fig. 6 need to be combined with a realistic treatment of BBOA partitioning for effective model inputs to describe BBOA evolution in the atmosphere.

## 5 Conclusions

We show that the OH-initiated aging of biomass burning emissions exhibit significant changes in BBOA composition and loading. These changes are dependent on OH exposure, and are especially large over the first few days after emission.

Significant amounts of SOA are formed from all fuels studied here, but SOA formation is highly variable. Despite large differences in fuel type and burning conditions, much of this variability can be explained by differences in the initial total NMOG concentration and OH exposure. Correlations between SOA formation and the concentration of initial measured NMOGs in the chamber at given OH exposures are good, with $r^2$ values between 0.64 and 0.83, and indicate SOA carbon yields between 24% (after 6 hours of equivalent atmospheric oxidation) to 56% (after 4 days). Given total NMOG

measurements from future field campaigns, the calculated SOA carbon yields can be used to estimate gross SOA formation from biomass burning in chemical transport models. However, these estimates would likely need to be used in conjunction with estimates of BBOA evaporation rates to calculate the net effect of aging on OA concentrations. Future work investigating the evolution of biomass burning emissions should attempt to further constrain the rates of BBOA evaporation and compare





the relative rates of oxidation and dilution from field and laboratory studies. In addition to this, laboratory studies on a wider range of fuels (i.e., those found in areas other than the western U.S.), and under a wider range of reaction conditions (i.e., different $RO_2$ reaction pathways), will help improve the ability to predict the loadings, properties, and impacts of biomass burning emissions globally.

## 5 Data availability

Data are available from the CSD NOAA archive at:

https://esrl.noaa.gov/csd/groups/csd7/measurements/2016firex/FireLab/DataDownload/ (NOAA, 2019)

## Author contribution

Data were interpreted and manuscript was written by CYL and JHK. Mini-chamber construction and operation were by CYL,
DHH, and CDC. AMS was operated and data were analyzed by CYL. PTR-ToF-MS was operated and data were analyzed by MMC, ARK, and KS. Experiments were conceived by JHK, CDC, and CW. All co-authors provided manuscript feedback and comments.

## Acknowledgments

This work was supported by NOAA AC4 award NA16OAR4310112. CYL and ARK were supported by the NSF graduate
research fellowship program. The authors would like to thank Colette Heald for helpful comments, and Edward Fortner, Timothy Onasch, Berk Knighton, Robert Yokelson, the entire of the FIREX science team, and Missoula Fire Sciences Lab staff for support during the project.

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



**Figure 1**. Changes in OA mass and composition as a function of aging time, assuming an atmospheric [OH] of $1.5 \times 10^6$ molec cm$^{-3}$. Each line represents a separate aging experiment. (a) Increase in OA mass with oxidation, showing variable SOA across all experiments. (b) Elemental ratios (red: O/C, black: H/C). (c) Change in average OS$_C$; reference line shows average change due to heterogeneous oxidation of laboratory flow tube experiments for comparison (Kroll et al., 2015). (d) Normalized fraction of AMS organic signal due to fragment $C_2H_4O_2^+$, a common primary biomass burning OA tracer, indicative of levoglucosan and related compounds.




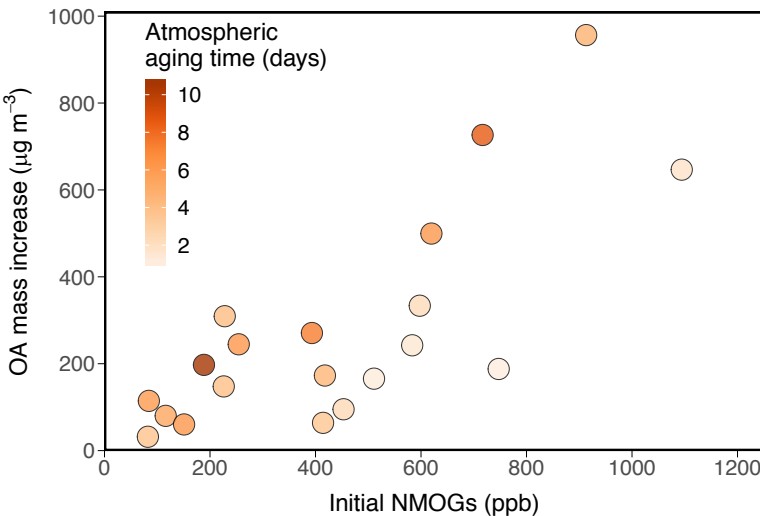

**Figure 2.** End-of-experiment SOA formation vs. total NMOG concentration in the chamber prior to OH oxidation. Points are colored by the atmospheric equivalent aging time corresponding to the end of each experiment.





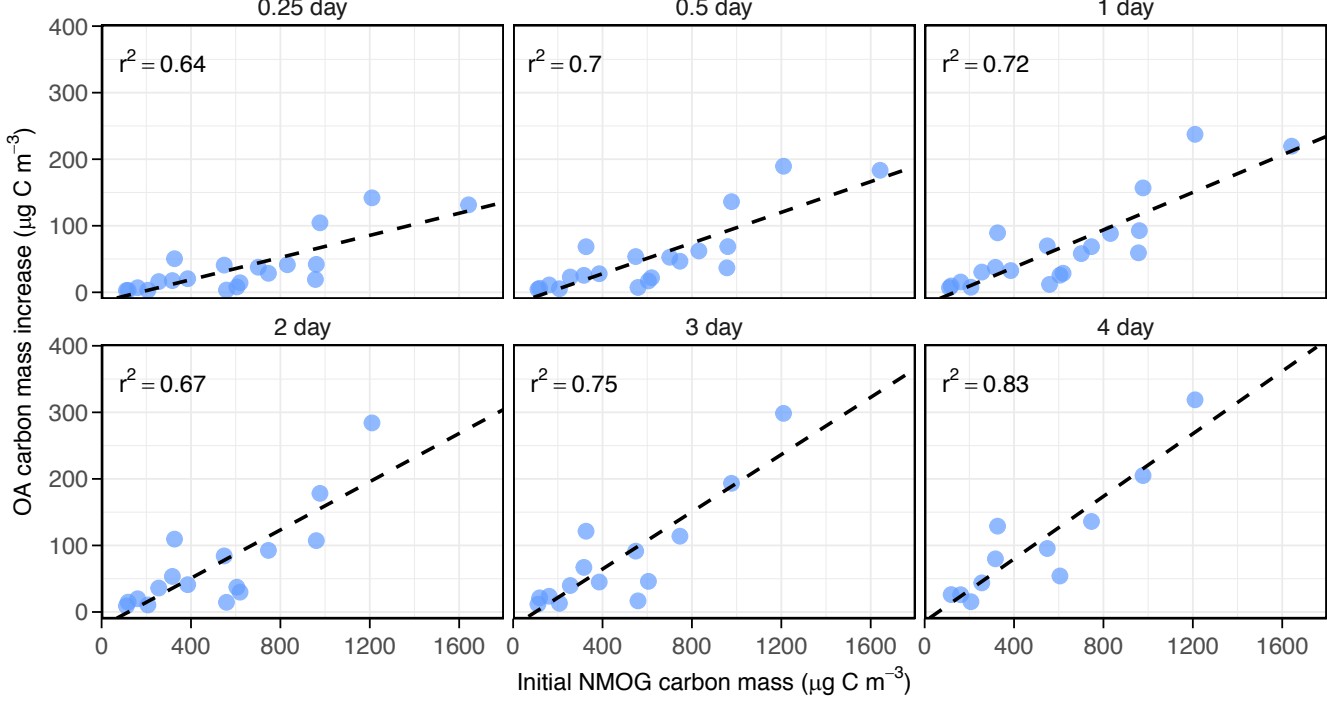

**Figure 3.** OA carbon mass added vs. initial NMOG carbon mass from PTR-ToF-MS measurements at various OH exposures (0.25 – 4 days of equivalent atmospheric aging). All subplots show correlation coefficients ($r^2$) of 0.64 or higher, and the linear relationships at longer aging times show larger slopes (i.e., more SOA; see Fig. 5).



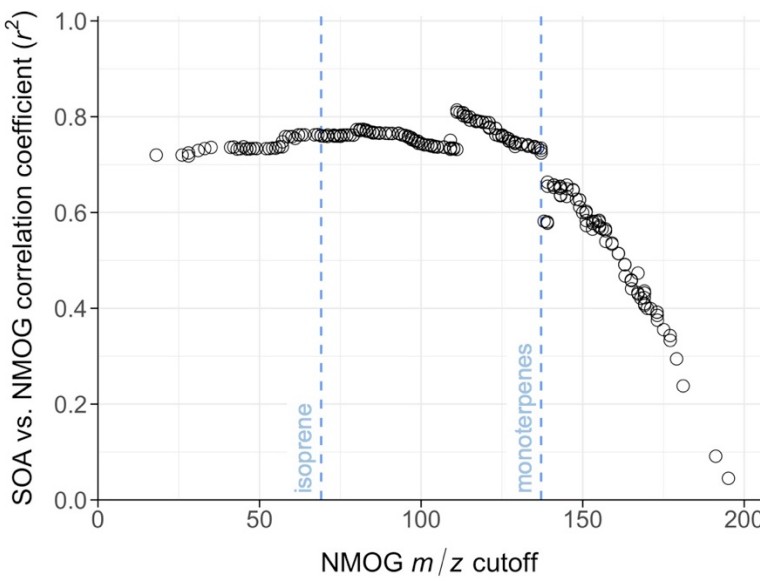

**Figure 4.** Correlation coefficients ($r^2$) between OA carbon mass added and the summed NMOG carbon mass above some ion mass ($m/z$) cutoff. For example, the point labeled "isoprene" shows the correlation between SOA carbon mass and initial NMOG carbon mass for all measured ions with mass-to-charge ratio equal to or greater than that of isoprene (i.e., species of molecular weight 68 g/mol or higher). Initial NMOG carbon mass is calculated prior to oxidation and data points all correspond to $r^2$ values at 1 day of atmospheric aging time. Correlation coefficients are high for all cutoff points below monoterpenes, then drop off due to loss of signal and the importance of compounds with lower molecular weight to SOA formation.





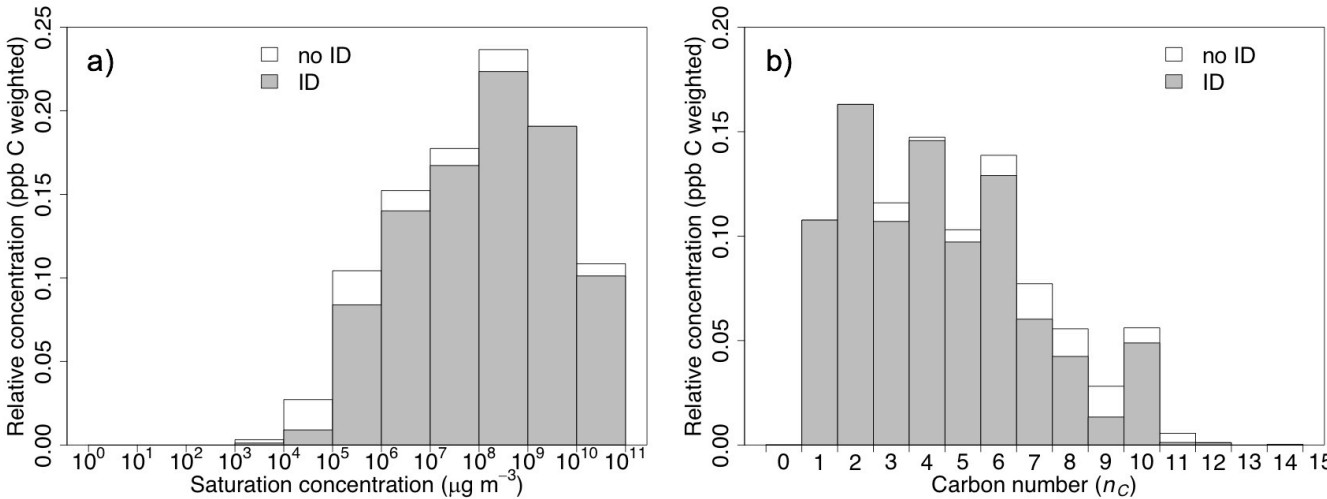

**Figure 5.** Estimated saturation vapor concentration distribution (a) and carbon number ($n_C$) distribution (b) for compounds (NMOGs) measured by the PTR-ToF-MS (Koss et al., 2018) in the chamber prior to oxidation, averaged over all burns. Distributions are separated into identified and unidentified ions and are weighted by ppb C.





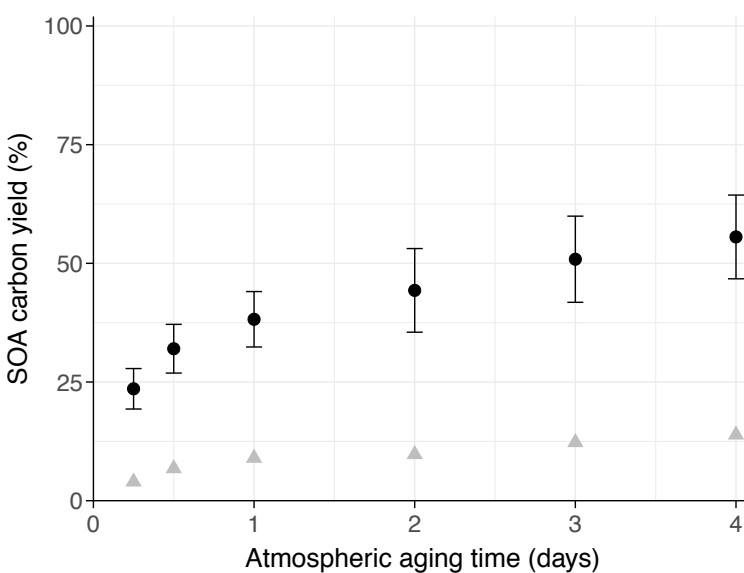

**Figure 6.** SOA carbon yield from aging of biomass burning emissions. Black points are carbon yields using our best estimate of OA carbon mass. Yields are calculated from the slopes of the linear relationships between SOA and initial NMOG by estimating the amount of measured NMOG carbon reacted at each respective time point using the average carbon-weighted OH rate coefficient for identified compounds and accounting for chamber dilution. Error bars are +/- 1σ in the slope of the linear fit between SOA and NMOG carbon mass. Grey triangles are estimated yields using AMS CE = 1 for all OA, both before and after aging, and is a common assumption made in previous studies on the aging of biomass burning. See Supporting Information (SI Table 2) for tabulated yields.