# Peer review of "Secondary organic aerosol formation from the laboratory oxidation of biomass burning emissions"

_Atmospheric Chemistry and Physics, 2019_

## Referee Comment (RC1) · Anonymous Referee #1 · 7 May 2019

The manuscript "Secondary organic aerosol formation from biomass burning emissions" presents the results of photooxidation studies of during the FIREX 2016 campaign, culminating in two pragmatic parameterizations that can be used to describe the organic aerosol change from aging biomass burning emissions. The first parameterization shows good correlations between the mass of non-methane organic gases to secondary organic aerosol yield for biomass burning emissions for a given OH exposure. This relationship is calculated by Christopher Lim et al. using aerosol mass spectrometer (AMS) organic aerosol mass measurements. These measurements are strongly influenced by the instrument collection efficiency (CE). The authors account for variability in the CE using the relationship between CE and mass fraction remaining after a thermodenuder. The second parameterization is the carbon-based SOA yield.

[Figure]

Both parameterizations will be useful to the community of chemical transport modelers, with some additional modeling for dilution. Given the potential importance and utility of these parameters, it would be prudent to include in the manuscript some additional details.

In the following, I will use P1.5 to indicate page 1, line 5 of the submitted manuscript.

Title: Accurate but not precise, in my opinion. Some reference could be made that this is a lab study and/or utilizes a photooxidation mini-chamber.

P4.20 Where/how is the additional O3 injected? Is there a sufficient jet to induce mixing? Is there any concern about a small area of highly concentrated O3 chemistry in the chamber? I'm curious about the O3/OH reactivity in general in your chamber, but more specifically at the injection port.

P5.2 Were seeded blanks (e.g. ammonium sulfate) ever run to establish a background OA production for this chamber and test the efficacy of the cleaning procedure?

P5.8 Did the black carbon measurements for these two methods agree? I wonder because the SP2 can saturate at high number concentrations, and the SP-AMS CE for BC requires some additional considerations (Ahern et al., 2016; Onasch et al., 2012; Willis et al., 2014.)

P5.26-30 I find your parameterization of CE very interesting and possibly broadly applicable. But why was it necessary at all? If you can calculate MFR, why not use the SEMS-measured size distribution to correct for CE directly? Given the large amount of variability in your FigS3, it is not obvious that using the correlation is an improvement in accuracy or precision over a size distribution correction.

Additionally, I don't agree with the statement that there was no good internal standard available. While I think that your CE parameterization could be very useful, it warrants verification by looking at other measurements more closely. For example, you state that there was an SO2 monitor, which would allow for a sulfur mass balance. Black

carbon, measured by SP-AMS or SP2, was present at useful concentrations for some of the experiments.

P5.31 Would you please confirm that the experiments used to calculate your CE were devoid of nucleation, particles grown outside the SEMS or AMS transmission/measurement ranges, and weren't unduly influenced by rBC after thermodenuding? Also, out of curiosity, how frequently was the thermodenuder valve switched, and therefore a new CE able to be calculated?

P6.2-3 Please provide a citation or clarify regarding the relationship between volatility and phase. It might be easier to provide citations that claim that SOA has been observed to be an amorphous solid with low volatility, and therefore is likely to bounce.

P8.5 What are the possible implications for the chamber OA concentration having decreased by two orders of magnitude from the beginning to the end of the experiment (FigS4)? It stands to reason that fewer of the semi-volatile SOA products will condense at low OA concentrations late in the experiment, but that any that do condense will have a larger impact on OSc.

P10.18 How does this approach compare with the measured PTR-MS NMOG concentrations? It's not obvious to me why you compare [calculated VOC reacted]/[measured SOA formed] instead of [measured VOC reacted]/[measured SOA formed]? Or to go backwards, can you use your [measured NMOG reacted]* this calculated SOA yield to predict SOA formation?

P11.19 Given the importance of dilution and volatility on the results presented here, is there information on the volatility of the POA and the SOA from the thermodenuder measurements, that can be compared to past campaigns?

Minor technical corrections: P3.22 "subalpine fir" rather than "subalpine fire" P7.25 Please include Hennigan et al. 2011 P11.20 Missing a period. FigS3 No red exponential fit FigS6 It looks like some plots have multiple y-values for a given time; some

boomerangs at 0-5 days where they should be smooth functions. Is it possible that the d-butanol injection is also being plotted? Does this change the OA enhancement ratios?

Ahern, A. T., Subramanian, R., Saliba, G., Lipsky, E. M., Donahue, N. M., and Sullivan, R. C.: Effect of secondary organic aerosol coating thickness on the real-time detection and characterization of biomass-burning soot by two particle mass spectrometers, Atmos. Meas. Tech., 9, 6117-6137, https://doi.org/10.5194/amt-9-6117-2016, 2016.

T. B. Onasch, A. Trimborn, E. C. Fortner, J. T. Jayne, G. L. Kok, L. R. Williams, P. Davidovits & D. R. Worsnop (2012) Soot Particle Aerosol Mass Spectrometer: Development, Validation, and Initial Application, Aerosol Science and Technology, 46:7, 804-817, DOI: 10.1080/02786826.2012.663948

Willis, M. D., Lee, A. K. Y., Onasch, T. B., Fortner, E. C., Williams, L. R., Lambe, A. T., Worsnop, D. R., and Abbatt, J. P. D.: Collection efficiency of the soot-particle aerosol mass spectrometer (SP-AMS) for internally mixed particulate black carbon, Atmos. Meas. Tech., 7, 4507-4516, https://doi.org/10.5194/amt-7-4507-2014, 2014.

---

## Referee Comment (RC2) · Anonymous Referee #2 · 13 Jun 2019

The Lim et al. manuscripts reports on SOA formation and aging in a series of chamber studies, conducted as part of FIREX. The chamber experiments were conducted at the Fire Sciences Laboratory, and biomass burning emissions provided the precursors for SOA formation; aging proceeded largely by exposure to OH. Gases and particles were characterized throughout the batch-mode experiments using a suite of instrumentation. Mass loadings, OA enhancement ratios, aging time (OH only), and bulk elemental composition are also reported for each experiment. One of the most significant findings of this work is that that AMS collection efficiencies may have been significantly underestimated in a wide range of published works, which affects interpretation of results and ultimately our understanding of SOA formation as gained from those works. The authors report that while OA enhancement ratios initially appeared

to be significantly higher than those reported in other publications, once aging time and collection efficiency were taken into account, the results were broadly consistent. The authors present a parameterization for SOA formation from biomass burning as a function of total NMOG and aging time that may prove useful when insufficient data regarding NMOG speciation and associated SOA yields are available. The paper is very well written, and the methodologies and results are clearly presented. It is an interesting and thought provoking paper, and likely will create much discussion and future research within the atmospheric community. Some specific comments and questions are provided below. Only minor edits are suggested before publication.

Scientific On p. 4 potential loses in the sampling line are discussed, and a figure is provided in the supplement illustrating the difference between gaseous emissions sampled in the stack directly, to those in the community inlet, binned by saturation concentration. The binned comparison does not suggest a systematic loss in the community inlet either across bins or as a function of volatility. However, there is a significant difference in one of the bins (C* 107-108). Is this difference well understood (e.g., likely due to a specific class of compounds)? And, how might this difference affect the results and analysis?

On p. 7, line 4 the authors state that the dilution factor prior to oxidation influenced observed initial aerosol mass and reference Table 1 in the supplement. It is not clear how the dilution factor is represented in the table. Is it a function of sampling time? This needs a bit more clarification/explanation.

The relationship between reported enhancement ratios in this work with previously published ratios is discussed on p. 7, first paragraph. The authors suggest that once aging and collection efficiency are taken into account, the results are broadly consistent with other results, and not overestimates. However, based on Fig. S6 panel b, where OA enhancement ratios are plotted as a function of aging time, the reported enhancements are still a factor of 2+ higher than Ortega et al. Are the Lim et al. enhancement ratios in the right panel actually for CE =1? They seem significantly higher than what is presented in the left panel (and it is assumed that both are for CE = 1). In addition, the average appears to be ∼3, which is the CE corrected-value reported in the main text.

In the discussion of correlation of measured SOA formation with different parameters (pp. 8-9), it is suggested that the reasonable correlations exist between SOA and total NMOG, with some relationship to POA, and when corrected for aging time. On p. 9, line 2, the authors state that the NMOG correlated well with POA (not shown), and thus POA is also well correlated with SOA. The correlation with POA, at least based on r2 values, actually appears to be better than the correlation with total NMOG. If the goal is a simplified parameterization, why not just use POA?

In the discussion of the results presented in Figs. 4 and 5, it is suggested that the relative insensitivity of SOA to NMOG m/z cutoff below ∼m/z 135 is either due to the contribution of higher volatility/lower molecular weight species, or other compounds which are not measured by the PTRMS, but correlated with these smaller NMOG molecules. It seems like there may be sufficient data available to look at the ratio of likely SOA precursors to total NMOG to test this hypothesis. Also, while it is not discussed, to the observed carbon yields themselves lend some insight to the likely precursors?

Editorial p. 8, line 20: Recommend changing "common SOA precursors" to "monoterpenes", since that is the only precursors show in S.7. p.9, line 23: It would be interesting to see the correlation plots of the two different temperature NMOG factors with SOA. Could these be added to the supplement?

---

## Author Response (AR1)

**Response to referee comments for:**

**Lim et al.**

**"Secondary organic aerosol formation from the laboratory oxidation of biomass burning emissions"**

**Referee #1**

Referee #1: Title: Accurate but not precise, in my opinion. Some reference could be made that this is a lab study and/or utilizes a photooxidation mini-chamber.

Author response: Title changed to "Secondary organic aerosol formation from the laboratory oxidation of biomass burning emissions" to better represent the work presented in the paper.

Referee #1: P4.20 Where/how is the additional O3 injected? Is there a sufficient jet to induce mixing? Is there any concern about a small area of highly concentrated O3 chemistry in the chamber? I'm curious about the O3/OH reactivity in general in your chamber, but more specifically at the injection port.

Author response: Ozone is generated with a pen-ray lamp external to the chamber and air carrying the $O_3$ is injected through a port separate from the injection port for the biomass burning emissions. The ozone concentration from the pen-ray lamp is about 1 ppm, but is mixed with humid air before injected into the chamber. The resulting $O_3$ concentration is approximately 200 – 300 ppb at the point of injection with a chamber mixing time of around 20 minutes. It is true that the area near the $O_3$ injection port will have higher $O_3$ concentrations relative to the rest of the bag which may lead to enhanced $O_3$ chemistry. However, from an experiment with no 254 nm UV, we do not observe strong changes in the composition of the aerosol with just $O_3$. Additionally, 254 UV makes OH in proportion to $O_3$, so the OH/$O_3$ ratio should not be sensitive to the $O_3$ concentration gradient in the chamber.

Referee #1: P5.2 Were seeded blanks (e.g. ammonium sulfate) ever run to establish a background OA production for this chamber and test the efficacy of the cleaning procedure?

Author response: Yes, seeded blanks (with ammonium sulfate particles) were run and showed negligible OA formation indicating that the cleaning procedure was successful. Deuterated butanol was not injected for the blanks (i.e., no OH exposure could be calculated), but $O_3$ and UV conditions were similar to those used in biomass burning oxidation experiments. The following text was added to the main manuscript (page 5, lines 4-5):

*"Seeded blanks (with ammonium sulfate particles) were run and showed negligible OA formation indicating that the cleaning procedure was successful."*

Referee #1: P5.8 Did the black carbon measurements for these two methods agree? I wonder because the SP2 can saturate at high number concentrations, and the SP-AMS CE for BC requires some additional considerations (Ahern et al., 2016; Onasch et al., 2012; Willis et al., 2014.)

Author response: SP2 and SP-AMS black carbon mass loadings generally agree within a factor of two, with SP2 mass loadings consistently higher than the measurements from the SP-AMS and a reasonably linear relationship ($r^2 = 0.9$). BC measurements from the SP2 are those reported in the manuscript. The SP2 was operated with time-varying flow rates to account for the large dynamic range in the BC concentrations over the course of an experiment. In this manner, coincidence and under-counting (saturation) issues were avoided. The SP2 was calibrated with size-selected fullerene soot, and mass concentrations were corrected for "missing mass" outside of the SP2 detection window via multi-modal log-normal fitting. However we note that SP-AMS BC measurements were used only to determine which experiments to filter out of the analysis (due to enhanced wall loss), and so any CE-related errors will not affect the results presented. Black carbon measurements and other details about primary emissions will be discussed further in a future publication (Cappa et al., in preparation).

Referee #1: P5.26-30 I find your parameterization of CE very interesting and possibly broadly applicable. But why was it necessary at all? If you can calculate MFR, why not use the SEMS-measured size distribution to correct for CE directly? Given the large amount of variability in your FigS3, it is not obvious that using the correlation is an improvement in accuracy or precision over a size distribution correction. Additionally, I don't agree with the statement that there was no good internal standard available. While I think that your CE parameterization could be very useful, it warrants verification by looking at other measurements more closely. For example, you state that there was an SO2 monitor, which would allow for a sulfur mass balance. Black carbon, measured by SP-AMS or SP2, was present at useful concentrations for some of the experiments.

Author response: While an SO₂ monitor was present, we observed that the measurements were unreliable due to, most likely, strong interferences from large concentrations of PAH's and other molecules, despite the internal scrubber in the SO₂ monitor. Further, it is not entirely clear to us how sulfur balance (gas + particle) would provide clear insights into the particle collection efficiency; we did not have an independent particulate sulfate measurement, but a gas-phase sulfur measurement. Given that the SO₂ measurements were compromised by interferences, we have now removed them from the list of measurements that were made. While we can use the SEMS-determined MFR, it is not clear how this would lead to a "direct" correction for the CE. It would provide an alternative approach, but with a complication that the MFR from the SEMS would include contributions from non-refractory material and thus does not directly address the issue of how the organic component of the particles responded to temperature changes. The CE changes observed derive, in part, from changes in the organic MFR that result from oxidation leading to less volatile OA. This would only partially be captured by the SEMS because of the contribution of non-refractory components (e.g. BC). BC cannot be used as an internal standard since the BC and organic mixing state varied dramatically between experiments, with some having the majority of the organic and BC being internally mixed (at low [OA]/[BC] ratios) and some having most of the organic material externally mixed from BC (at high [OA]/[BC] ratios) (McClure et al., *in prep.*). Given these overall issues, we believe that the organic MFR links more closely to the physical changes that occur. The text in the manuscript was amended (page 5, line 31):

*"However, we were unable to find a suitable tracer in these experiments: sulfate changes as a result of oxidation of emitted SO2, black carbon (when present in high concentrations) exhibited wall losses different from OA (as described below)* **and appeared not to be homogeneously mixed with the OA (McClure et al., in prep),** *and POA tracers (such as the C7H11+ ion, recently used by Ahern et al. (2019)) are likely to be lost via heterogeneous oxidation at the high OH exposures examined here. Thus, corrections for CE, dilution, and particle wall loss were carried out individually, as described below."*

Referee #1: P5.31 Would you please confirm that the experiments used to calculate your CE were devoid of nucleation, particles grown outside the SEMS or AMS transmission/measurement ranges, and weren't unduly influenced by rBC after thermodenuding? Also, out of curiosity, how frequently was the thermodenuder valve switched, and therefore a new CE able to be calculated?

Author response: Yes, only data that did not show significant nucleation and had low rBC loadings were used to calculate the CE parameterizations. In addition, only SEMS and PToF size distributions that could be fit to a lognormal function were used. The thermodenuder valve was switched from thermodenuder to bypass every two minutes. Some additional text clarifying these details is included in the main text (page 6, lines 4-6):

*"CE and particle density were calculated by comparing AMS particle time-of-flight (PToF) and SEMS size distributions (Bahreini et al., 2005) for a subset of data points with PToF and SEMS distributions that could be fit to lognormal functions, did not show significant particle nucleation, and had low rBC concentration (see below)."*

Referee #1: P6.2-3 Please provide a citation or clarify regarding the relationship between volatility and phase. It might be easier to provide citations that claim that SOA has been observed to be an amorphous solid with low volatility, and therefore is likely to bounce.

Author response: This is an excellent idea; we have included citations for Matthew et al. (2008) describing collection efficiencies as a function of particle phase in the AMS, as well as Virtanen et al. (2010) showing that SOA can be an amorphous solid.

Referee #1: P8.5 What are the possible implications for the chamber OA concentration having decreased by two orders of magnitude from the beginning to the end of the experiment (FigS4)? It stands to reason that fewer of the semi-volatile SOA products will condense at low OA concentrations late in the experiment, but that any that do condense will have a larger impact on OSc.

Author response: From Fig. S4, the organic concentration decreases by only one order of magnitude (200 μg/m$^3$ to 20 μg/m$^3$). Although semi-volatile gases are less likely to condense at low OA concentrations observed at the end of experiments due to significant dilution over the course of each experiment, most reactive gases are likely to have reacted relatively early in the experiment when dilution has less of an impact (see Fig. 6, most SOA growth occurs within ~2 days). Indeed, those gases (and potentially secondary products) that do react at longer OH exposures and condense are likely to have a large impact on the calculated average OS$_C$ and elemental ratios of the OA. Thus, the calculated OS$_C$ and elemental ratios may be more representative of the oxidized long-lived gases (which condense) rather than the BBOA + SOA as a whole. The following additional text was added to the manuscript to clarify this point (page 8, lines 7-8):

"*Over longer timescales, when dilution is more significant and OA concentrations are lower, calculated OS$_C$ are likely to reflect the oxidation of longer-lived gases.*"

Referee #1: P10.18 How does this approach compare with the measured PTR-MS NMOG concentrations? It's not obvious to me why you compare [calculated VOC reacted]/[measured SOA formed] instead of [measured VOC reacted]/[measured SOA formed]? Or to go backwards, can you use your [measured NMOG reacted]* this calculated SOA yield to predict SOA formation?

Author response: We are not able to directly measure the amount of NMOG reacted, due to the dilution loss of NMOG (gases removed from the chamber before they can be reacted with OH), formation of secondary gas-phase products, and potential off-gassing of non-SOA forming low molecular weight NMOG from the chamber walls, leading to calculated SOA yields greater than unity. Thus, only initial NMOG measurements are used for the analysis and reacted NMOG are calculated as described in the text. This explanation is now included in the main text (page, lines)

Referee #1: P11.19 Given the importance of dilution and volatility on the results presented here, is there information on the volatility of the POA and the SOA from the thermodenuder measurements, that can be compared to past campaigns?

Author response: Hennigan et al. (2011) observe an increase in organic MFR (80 C) for fires that show high OA enhancement as well as fires that show low OA enhancement. We do observe a decrease in volatility with oxidation – POA is relatively volatile, while SOA is less so. However, because we only used one temperature for the thermodenuder (250 C), we cannot obtain a volatility distribution as they do in May et al. (2013) where MFR was measured over a range of temperatures from 20 C to 120 C. Thus, comparing with either data set directly is difficult since our thermodenuder was run only at a single temperature that was much higher than the thermodenuder temperatures of the previously mentioned studies. Page 6, lines 8-9 were edited to mention that increases in MFR with oxidation are consistent with previous studies:

*"Generally, POA has low organic MFR (i.e., relatively volatile) and particle MFR increases with oxidation, consistent with previous work (Hennigan et al., 2011)."*

Hennigan, C J, Miracolo, M. A., Engelhart, G. J., May, A. A., Presto, A. A., Lee, T., & Sullivan, A. P. (2011). Chemical and physical transformations of organic aerosol from the photo-oxidation of open biomass burning emissions in an environmental chamber. *Atmospheric Chemistry and Physics*, 7669–7686. https://doi.org/10.5194/acp-11-7669-2011

May, Andrew A., Levin, E. J. T., Hennigan, C. J., Riipinen, I., Lee, T., Collett, J. L., et al. (2013). Gas-particle partitioning of primary organic aerosol emissions: 3. Biomass burning. *Journal of Geophysical Research Atmospheres*, 118(19), 11327–11338. https://doi.org/10.1002/jgrd.50828

Minor technical corrections:

Referee #1: P3.22 "subalpine fir" rather than "subalpine fire"

Author response: Corrected.

Referee #1: P7.25 Please include Hennigan et al. 2011

Author response: Citation now included.

Referee #1: P11.20 Missing a period.

Author response: Corrected.

Referee #1: FigS3 No red exponential fit

Author response: Red exponential fit was from an old version of the plot. Text referring to it has been removed.

Referee #1: FigS6 It looks like some plots have multiple y-values for a given time; some boomerangs at 0-5 days where they should be smooth functions. Is it possible that the d-butanol injection is also being plotted? Does this change the OA enhancement ratios?

Author response: D-butanol injection is not being plotted. The left panel of Fig. S6 is actually in units of hours to compare with Hennigan et al. (2011), not days as in the right panel. The multiple y-values for given aging times is due to noise in the OH exposure measurement at very low OH exposures corresponding to the first few minutes of the chamber experiment. The caption on the figure has been edited to describe this effect.

Ahern, A. T., Subramanian, R., Saliba, G., Lipsky, E. M., Donahue, N. M., and Sullivan, R. C.: Effect of secondary organic aerosol coating thickness on the real-time detection and characterization of biomass-burning soot by two particle mass spectrometers, Atmos. Meas. Tech., 9, 6117-6137, https://doi.org/10.5194/amt-9-6117-2016, 2016.

T. B. Onasch, A. Trimborn, E. C. Fortner, J. T. Jayne, G. L. Kok, L. R. Williams, P. Davidovits & D. R. Worsnop (2012) Soot Particle Aerosol Mass Spectrometer: Development, Validation, and Initial Application, Aerosol Science and Technology, 46:7, 804-817, DOI: 10.1080/02786826.2012.663948 Willis, M. D., Lee, A. K. Y., Onasch, T. B., Fortner, E. C.,

Williams, L. R., Lambe, A. T., Worsnop, D. R., and Abbatt, J. P. D.: Collection efficiency of the soot-particle aerosol mass
spectrometer (SP-AMS) for internally mixed particulate black carbon, Atmos. Meas. Tech., 7, 4507-4516, https://doi.org/10.5194/amt-7-4507-2014, 2014.

**Referee #2**

Scientific

Referee #2: On p. 4 potential loses in the sampling line are discussed, and a figure is provided in the supplement illustrating
the difference between gaseous emissions sampled in the stack directly, to those in the community inlet, binned by saturation concentration. The binned comparison does not suggest a systematic loss in the community inlet either across bins or as a function of volatility. However, there is a significant difference in one of the bins (C* $10^7$-$10^8$). Is this difference well understood (e.g., likely due to a specific class of compounds)? And, how might this difference affect the results and analysis?

Author response: The compounds in this bin do not correspond to a specific class of compounds but to a variety of compounds, including: acetic acid, furfural, furanone, monoterpenes, and methyl glyoxal/acrylic acid (in order of decreasing average abundance across all experiments). The presence of well-known SOA precursors (e.g., monoterpenes) in this volatility bin suggest that it is possible that some SOA precursors are in lower abundance in the mini-chamber relative to the stack. Another possibility is that acetic acid, the most abundant compound in this volatility bin, is preferentially lost during transport to the chamber due to accumulated water on the surfaces of the community inlet. Although preferential loss of some compounds in this volatility bin may affect SOA yields, the observation that total NMOGs correlates well with SOA suggests that some preferential loss of a small subset of compounds would not greatly affect SOA formation.

Referee #2: On p. 7, line 4 the authors state that the dilution factor prior to oxidation influenced observed initial aerosol mass and reference Table 1 in the supplement. It is not clear how the dilution factor is represented in the table. Is it a function of sampling time? This needs a bit more clarification/explanation.

Author response: This text simply refers to the fact that after sampling, experiments are diluted by varying amounts depending on the length of time between sampling and initiation of oxidation (254 nm UV lights). The sentence has been reworded for clarity (page 7, line 8-9):

*"The total, initial aerosol mass in the chamber varied widely from experiment to experiment (SI Table 1), averaging 130 ± 103 $\mu$g m$^{-3}$ (mean ±1$\sigma$), depending on the amount of fuel burned, fuel type, sampling time, and dilution prior to oxidation."*

Referee #2: The relationship between reported enhancement ratios in this work with previously published ratios is discussed on p. 7, first paragraph. The authors suggest that once aging and collection efficiency are taken into account, the results are broadly consistent with other results, and not overestimates. However, based on Fig. S6 panel b, where OA enhancement ratios are plotted as a function of aging time, the reported enhancements are still a factor of 2+ higher than Ortega et al. Are the Lim et al. enhancement ratios in the right panel actually for CE =1? They seem significantly higher than what is presented in the left panel (and it is assumed that both are for CE = 1). In addition, the average appears to be ~3, which is the CE corrected-value reported in the main text.

Author response: We thank the reviewer for pointing this out; Fig. S6b was mistakenly showing the CE corrected data instead of CE = 1. The figure has been updated so that both panels show CE = 1; however, even after correcting the figure, OA enhancements measured in the mini-chamber are significantly higher than those seen in Ortega et al. (2013). The text in the manuscript has been changed to reflect this (page 7, lines 18-23):

*"The average OA enhancement ratio was 3.5 ± 1.7. This is considerably higher than reported in previous studies (Ahern et al., 2019; Hennigan et al., 2011; Ortega et al., 2016; Tkacik et al., 2017), but once differences in OH exposure (as well as AMS CE) are taken into account, these results are broadly consistent with previous chamber studies and only somewhat higher than previous flow tube experiments (Fig. S6)."*

Referee #2: In the discussion of correlation of measured SOA formation with different parameters (pp. 8-9), it is suggested that the reasonable correlations exist between SOA and total NMOG, with some relationship to POA, and when corrected for aging time. On p. 9, line 2, the authors state that the NMOG correlated well with POA (not shown), and thus POA is also well correlated with SOA. The correlation with POA, at least based on r2 values, actually appears to be better than the correlation with total NMOG. If the goal is a simplified parameterization, why not just use POA?

Author response: While we provide a simple SOA parameterization based on NMOG (i.e., carbon yield from biomass burning emission), the goal of the paper is a more mechanistic understanding of the underlying chemistry and gas-phase precursors. As such, we present the NMOG parameterization in the manuscript, but also provide the relationships between SOA and POA Fig. S9 for purposes where POA measurements may be more readily available. The following text was added to the main text (page 9, lines 12-14):

*"As the goal of this work is to provide a more mechanistic understanding of the underlying chemistry, relationships between SOA and NMOGs are shown in the main text; relationships between POA and SOA are given in Fig. S9 for purposes where POA measurements may be more readily available."*

Referee #2: In the discussion of the results presented in Figs. 4 and 5, it is suggested that the relative insensitivity of SOA to NMOG m/z cutoff below ~m/z 135 is either due to the contribution of higher volatility/lower molecular weight species, or other compounds which are not measured by the PTRMS, but correlated with these smaller NMOG molecules. It seems like there may be sufficient data available to look at the ratio of likely SOA precursors to total NMOG to test this hypothesis. Also, while it is not discussed, to the observed carbon yields themselves lend some insight to the likely precursors?

Author response: We appreciate this suggestion. We have looked in more detail at the precursors and their relationships (or lack thereof) with SOA formation. We were not able to determine specific precursors or a specific class of compounds that show significantly stronger correlations with SOA than the total NMOG loading. Monoterpenes, low/high temperature factors, each m/z cutoff, and compounds binned by carbon number or volatility did not show improved correlations with SOA compared to total NMOG loadings. In addition, we also examined the correlation between SOA and the top SOA precursors identified in Bruns et al. (2016), including benzene, phenol, naphthalene, and related compounds and they do not show good correlations SOA formation. These compounds have are now explicitly mentioned in the main text (page 9, line 30).

Referee #2: p. 8, line 20: Recommend changing "common SOA precursors" to "monoterpenes", since that is the only precursors show in S.7.

Author response: Text changed to "monoterpenes."

Referee #2: p.9, line 23: It would be interesting to see the correlation plots of the two different temperature NMOG factors with SOA. Could these be added to the supplement?

Author response: SOA (carbon mass) correlation plots for low and high temperature combustion factors have been added to the supplement (Fig. S10). High and low temperature factors for each burn are calculated by calculating the fraction high/low factor for the initial NMOG composition (before oxidation) and multiplying by the total NMOG loading (ppb C). Reference to SI added to the main text (page 9, line 34 – page 10, line 2):

[revised manuscript text omitted]

Yuan, B., Koss, A. R., Warneke, C., Coggon, M., Sekimoto, K., & De Gouw, J. A. (2017). Proton-Transfer-Reaction Mass Spectrometry: Applications in Atmospheric Sciences. *Chemical Reviews*, *117*(21), 13187–13229. https://doi.org/10.1021/acs.chemrev.7b00325

[Figure]

**Figure 1**. Changes in OA mass and composition as a function of aging time, assuming an atmospheric [OH] of $1.5 \times 10^6$ molec cm$^{-3}$. Each line represents a separate aging experiment. (a) Increase in OA mass with oxidation, showing variable SOA across all experiments. (b) Elemental ratios (red: O/C, black: H/C). (c) Change in OS$_C$; reference line shows average change due to heterogeneous oxidation of laboratory flow tube experiments for comparison (Kroll et al., 2015). (d) Normalized fraction of AMS organic signal due to fragment $C_2H_4O_2^+$, a common primary biomass burning OA tracer, indicative of levoglucosan and related compounds.

[Figure]

**Figure 2.** End-of-experiment SOA formation vs. total NMOG concentration in the chamber prior to OH oxidation. Points are colored by the atmospheric equivalent aging time corresponding to the end of each experiment.

[Figure]

**Figure 3.** OA carbon mass added vs. initial NMOG carbon mass from PTR-ToF-MS measurements at various OH exposures (0.25 – 4 days of equivalent atmospheric aging). All subplots show correlation coefficients ($r^2$) of 0.64 or higher, and the linear relationships at longer aging times show larger slopes (i.e., more SOA; see Fig. 5).

[Figure]

**Figure 4.** Correlation coefficients ($r^2$) between OA carbon mass added and the summed NMOG carbon mass above some ion mass ($m/z$) cutoff. For example, the point labeled "isoprene" shows the correlation between SOA carbon mass and initial NMOG carbon mass for all measured ions with mass-to-charge ratio equal to or greater than that of isoprene (i.e., species of molecular weight 68 g/mol or higher). Initial

NMOG carbon mass is calculated prior to oxidation and data points all correspond to $r^2$ values at 1 day of atmospheric aging time. Correlation coefficients are high for all cutoff points below monoterpenes, then drop off due to loss of signal and the importance of compounds with lower molecular weight to SOA formation.

[Figure]

**Figure 5.** Estimated saturation vapor concentration distribution (a) and carbon number ($n_C$) distribution (b) for compounds (NMOGs) measured by the PTR-ToF-MS (Koss et al., 2018) in the chamber prior to oxidation, averaged over all burns. Distributions are separated into identified and unidentified ions and are weighted by ppb C.

[Figure]

**Figure 6.** SOA carbon yield from aging of biomass burning emissions. Black points are carbon yields using our best estimate of OA carbon mass. Yields are calculated from the slopes of the linear relationships between SOA and initial NMOG by estimating the amount of measured NMOG carbon reacted at each respective time point using the average carbon-weighted OH rate coefficient for identified compounds and accounting for chamber dilution. Error bars are +/- 1σ in the slope of the linear fit between SOA and NMOG carbon mass. Grey triangles are estimated yields using AMS CE = 1 for all OA, both before and after aging, and is a common assumption made in previous studies on the aging of biomass burning. See Supporting Information (SI Table 2) for tabulated yields.

---

## Referee Report (RR1)

This work represents a significant contribution to the understanding of SOA production from biomass burning emissions. Fig S6 shows that reported SOA production are consistent with what has been previously reported, with the principal difference being the use of a new parameterization for AMS collection efficiency. As such, the new CE parameterization should be more rigorously verified. I will respond to comments in-line below.

Referee #1: P5.8 Did the black carbon measurements for these two methods agree? I wonder because the SP2 can saturate at high number concentrations, and the SP-AMS CE for BC requires some additional considerations (Ahern et al., 2016; Onasch et al., 2012; Willis et al., 2014.)

Author response: SP2 and SP-AMS black carbon mass loadings generally agree within a factor of two, with SP2 mass loadings consistently higher than the measurements from the SP-AMS and a reasonably linear relationship ($r2$ = 0.9). BC measurements from the SP2 are those reported in the manuscript. The SP2 was operated with time-varying flow rates to account for the large dynamic range in the BC concentrations over the course of an experiment. In this manner, coincidence and under-counting (saturation) issues were avoided. The SP2 was calibrated with size-selected fullerene soot, and mass concentrations were corrected for "missing mass" outside of the SP2 detection window via multi-modal log-normal fitting. However we note that SP-AMS BC measurements were used only to determine which experiments to filter out of the analysis (due to enhanced wall loss), and so any CE-related errors will not affect the results presented. Black carbon measurements and other details about primary emissions will be discussed further in a future publication (Cappa et al., in preparation).

Please clarify which BC measurements are used when in the manuscript.

Referee #1: P5.26-30 I find your parameterization of CE very interesting and possibly broadly applicable. But why was it necessary at all? If you can calculate MFR, why not use the SEMS-measured size distribution to correct for CE directly? Given the large amount of variability in your FigS3, it is not obvious that using the correlation is an improvement in accuracy or precision over a size distribution correction. Additionally, I don't agree with the statement that there was no good internal standard available. While I think that your CE parameterization could be very useful, it warrants verification by looking at other measurements more closely. For example, you state that there was an SO2 monitor, which would allow for a sulfur mass balance. Black carbon, measured by SP-AMS or SP2, was present at useful concentrations for some of the experiments.

Author response: While an SO2 monitor was present, we observed that the measurements were unreliable due to, most likely, strong interferences from large concentrations of PAH's and other molecules, despite the internal scrubber in the SO2 monitor. Further, it is not entirely clear to us how sulfur balance (gas + particle) would provide clear insights into the particle collection efficiency; we did not have an independent particulate sulfate measurement, but a gas-phase sulfur measurement. Given that the SO2 measurements were compromised by interferences, we have now removed them from the list of measurements that were made.

If any of your $SO_2$ measurements were not contaminated, you could calculate the mass of $SO_2$ oxidized. That $SO_2$ would likely condense onto the particles as $(NH_4)_2SO_4$, which can be measured by your AMS. This would give you another way to verify that you understand your collection efficiency.

While we can use the SEMS-determined MFR, it is not clear how this would lead to a "direct" correction for the CE. It would provide an alternative approach, but with a complication that the MFR from the SEMS would include contributions from non-refractory material and thus does not directly address the issue of how the organic component of the particles responded to temperature changes.

To clarify, I am not proposing you use the SEMS-determined MFR. Rather, I question whether your parameterization of AMS CE is an improvement over using the SEMS-measured volume and AMS measured mass to calculate CE. It appears to me that given the degree of uncertainty in your CE parameterization (especially for low MFR), the contribution of rBC to SEMS-measured particle volume may be small in comparison. For most experiments, you would overestimate OA mass by less than 10%, 20% at the most for your selected experiments. That is assuming that you don't account for the measured BC mass (Slowik et al. 2004). I would also posit that it may not be necessary to fit a Gaussian to the particle size distributions, depending on how much variability you see in the calculated OA density.

This should be compared to the uncertainty in OA mass introduced by applying your parameterization. I do think your parameterization has merit and should be published, but it needs to be validated more rigorously and I'm not convinced it is the best way to calculate OA mass given this suite of instruments.

I would also be curious to know if the scatter in Fig S3 is due to precision of the measurement, or if it is due to different experiments having slightly different relationships between CE and MFR.

Jay G. Slowik, K. Stainken, Paul Davidovits, L. R. Williams, J. T. Jayne, C. E. Kolb, Douglas R. Worsnop, Y. Rudich, Peter F. DeCarlo & Jose L. Jimenez (2004) Particle Morphology and Density Characterization by Combined Mobility and Aerodynamic Diameter Measurements. Part 2: Application to Combustion-Generated Soot Aerosols as a Function of Fuel Equivalence Ratio, Aerosol Science and Technology, 38:12, 1206-1222, DOI: 10.1080/027868290903916

---

## Author Response (AR2)

Response to Referee Report #1:

Referee #1: This work represents a significant contribution to the understanding of SOA production from biomass burning emissions. Fig S6 shows that reported SOA production are consistent with what has been previously reported, with the principal difference being the use of a new parameterization for AMS collection efficiency. As such, the new CE parameterization should be more rigorously verified. I will respond to comments in-line below.

Author response: Fig. S6 actually shows some discrepancy between SOA from previous biomass burning oxidation experiments and SOA in our results. While our results generally agree with Hennigan et al. (2011) when assuming a constant AMS collection efficiency (CE) of 1, our SOA formation looks to be considerably higher than those from Ortega et al. (2013). To address the referee's comment that the CE parameterization should be better verified, we have added a comparison of SOA mass using our SEMS measurement to show that OA mass using the MFR CE parameterization agrees well with the OA mass derived from SEMS volume after accounting for density. In addition, we have clarified our description of our CE parameterization.

Referee #1: Please clarify which BC measurements are used when in the manuscript.

Author response: SP2 vs. SP-AMS BC measurements have been clarified throughout the text.

Referee #1: If any of your SO2 measurements were not contaminated, you could calculate the mass of SO2 oxidized. That SO2 would likely condense onto the particles as (NH4)2SO4, which can be measured by your AMS. This would give you another way to verify that you understand your collection efficiency.

Author response: It is unlikely that SO2 measurements are not contaminated due to the likely presence of interfering compounds like PAHs in most experiments. Additionally, variability in mixing state (internal vs. externally mixed particles) further complicates the use of sulfate as an aid to calculate AMS collection efficiency. Moreover, the sulfate grew in only after the SOA formation (due to the slower reaction rate with OH), providing little information on the evolving CE during SOA formation.

Referee #1: To clarify, I am not proposing you use the SEMS-determined MFR. Rather, I question whether your parameterization of AMS CE is an improvement over using the SEMS-measured volume and AMS measured mass to calculate CE. It appears to me that given the degree of uncertainty in your CE parameterization (especially for low MFR), the contribution of rBC to SEMS-measured particle volume may be small in comparison. For most experiments, you would overestimate OA mass by less than 10%, 20% at the most for your selected experiments. That is assuming that you don't account for the measured BC mass (Slowik et al. 2004). I would also posit that it may not be necessary to fit a Gaussian to the particle size distributions, depending on how much variability you see in the calculated OA density.

This should be compared to the uncertainty in OA mass introduced by applying your parameterization. I do think your parameterization has merit and should be published, but it needs to be validated more rigorously and I'm not convinced it is the best way to calculate OA mass given this suite of instruments.

I would also be curious to know if the scatter in Fig S3 is due to precision of the measurement, or if it is due to different experiments having slightly different relationships between CE and MFR.

Jay G. Slowik, K. Stainken, Paul Davidovits, L. R. Williams, J. T. Jayne, C. E. Kolb, Douglas R. Worsnop, Y. Rudich, Peter F. DeCarlo & Jose L. Jimenez (2004) Particle Morphology and Density Characterization by Combined Mobility and Aerodynamic Diameter Measurements. Part 2: Application to Combustion-

Generated Soot Aerosols as a Function of Fuel Equivalence Ratio, Aerosol Science and Technology, 38:12, 1206-1222, DOI: 10.1080/027868290903916

Author response: We thank the reviewer for the clarification. The scatter in Fig. S3 is both due to uncertainty in the measurement and slight differences with the relationship between CE and MFR from different experiments (this explanation has been added to the caption of Fig. S3). Due to the significant increases in OA density that we observe with oxidation, we must be able to fit the SEMS and AMS PToF size distributions to Gaussian distributions in order to accurately estimate mass. To clarify, our MFR CE parameterization does in fact use SEMS volume multiplied by density compared to AMS organic mass to calculate CE, which is essentially what the referee suggests. This has been clarified in the main text (page 6, lines 4-5):

"CE and particle density were calculated by comparing AMS particle time-of-flight (PToF) and SEMS size distributions along with AMS organic mass and SEMS volume (Bahreini et al., 2005)."

The MFR-based parameterization is necessary because of the sparsity of data points for which OA density (and CE) can be calculated with any precision (since signal-to-noise is limited for most experiments). To confirm that the parameterization used in the main manuscript provides a reasonable estimate of OA mass, we have calculated the SEMS-volume-derived OA mass after accounting for density and directly compared this with our OA mass calculated using the MFR CE parameterization. As shown below, the correlation between these two OA mass calculations is good ($r^2 = 0.90$) with a slope of 0.85. This figure has been added to the Supplementary Material (Fig. S11) and is referenced in the main text (page 6, lines 8-10):

"An alternative calculation of OA mass using SEMS volume multiplied by OA density agrees with OA mass calculated using this MFR CE parameterization and is shown in Fig. S11."

In addition, we have added additional text referring to a recent review on biomass burning SOA that discusses AMS CE briefly (page 11, lines 11-14):

[revised manuscript text omitted]

**S4. OA wall loss fit**

[Figure]

**Figure S4.** Wall loss fit for dark experiment (Fire 63). Wall loss time constant equals 35 minutes, based on fit of dilution-corrected OA mass.

**S5. Dilution corrected primary IVOCs**

[Figure]

**Figure S5.** Time series for dilution-corrected, high molecular weight gas phase compounds measured by PTR-ToF-MS. Dilution corrected concentrations are stable, indicating the impact of vapor wall loss for these compounds is not a major loss process over the timescales of these experiments.

**S6. Comparison between previous FSL aging studies**

[Figure]

**Figure S6.** Comparison between OA enhancement ratios for this work and previous Fire Lab aging studies. Panel on left is comparison to room-burn, large chamber oxidation from Hennigan et al. (2011). Recent work from Ahern et al. (2019) is also roughly consistent with this, but includes an internal CE correction. Panel on right is comparison to room-burn, flow tube oxidation from Ortega et al. (2013). Data are not corrected for AMS collection efficiency in order to compare with published work (CE = 1). Multiple y-values for given aging times on the left-hand panel are due to noise in the OH exposure measurement at very low OH exposures, corresponding to the first few minutes of the chamber experiment.

**S7. OA enhancement ratio scatterplots**

[Figure]

**Figure S7.** Scatterplots of OA enhancement ratio (end of experiment) vs. various parameters: (a) OH exposure, (b) POA mass, (c) monoterpene concentration, (d) total PTR species. No single parameter shows a strong relationship with OA enhancement ratio.

**S8. Effect of aerosol loading on carbon yield**

[Figure]

**Figure S8.** OA carbon mass added as a function of initial PTR signal and colored by OA mass in suspended in the chamber (1 day of equivalent exposure). Higher aerosol mass in the chamber contributes to higher conversion of gas-phase carbon to SOA.

**S9. POA vs. SOA scatterplots**

[Figure]

**Figure S9.** Scatterplots of SOA mass vs. POA mass at OH exposures equivalent to 0.25 days of atmospheric aging to 4 days of aging. Dashed lines are linear regressions described by the fit equations.

**S10. SOA vs. gas-phase high and low temperature factors**

[Figure]

**Figure S10.** OA carbon mass formed (1-day equivalent atmospheric aging) vs. low- and high-temperature factor loadings (ppb C). Initial chamber NMOG composition was fit to low and high temperature factors described in Sekimoto et al. (2018) to calculate factor fractions for each fire. Low- and high-temperature factor fractions were then multiplied by total NMOG concentration (ppb C) prior to oxidation to determine factor loadings. High- and low- temperature factor loadings show poor correlations with SOA carbon.

**S11. OA mass from MFR CE parameterization compared with mass directly from SEMS volume and density**

> **Commented [CL1]:** Added figure

[Figure]

**Fig. S11.** Comparison of calculated OA mass using the MFR CE parameterization (main text) and OA mass calculation directly using SEMS volume multiplied by OA density. OA mass calculated using the parameterization is slightly greater than mass calculated directly from SEMS volume and density (slope = 0.85) due to exclusion of BC and inorganic mass, but estimates agree well with each other ($r^2$ = 0.90). The small discrepancy between OA mass calculations does not affect the overall conclusions of the study.